# Epithelial changes of congenital intestinal obstruction in a rat model

Quentin Ballouhey[1,2]*, Laurent Fourcade[1,2], Laurence Richard[1,3], Camille Bellet[1], Chaharazed El Hamel[5], Jean Michel Vallat[1,3], Franck Sturtz[1,4], Sylvie Bourthoumieu[1,5]*

1 Myelin Maintenance and Peripheral Neuropathies, EA6309, University of Limoges, Limoges, France,
2 Department of Pediatric Surgery, UHC Limoges, Limoges, France, 3 Department of Neurology, UHC Limoges, Limoges, France, 4 Department of Biochemistry and Molecular Genetics, Centre Hospitalier Universitaire de Limoges, Limoges, France, 5 Department of Histology, Cytology, and Cytogenetics, Centre Hospitalier Universitaire de Limoges, Limoges, France

☯ These authors contributed equally to this work.
* qballouhey@gmail.com (QB); sylvie.bourthoumieu@unilim.fr (SB)

## Abstract

### Introduction

Intestinal atresia is a rare congenital affliction that is often associated with severe bacterial infections despite adequate neonatal surgery. Previous studies have focused on enteric nervous system variations. We hypothesized that epithelial systems (ES) may also be involved in the pathophysiology of postnatal disorders.

### Materials and methods

Global gene expression was measured by transcriptomic analysis in a rat model of induced intestinal atresia. The analyses then focused on genes involved in ES (enterocytes and goblet cells). Rat fetus small intestines at various stages of development (ED15, ED17, ED19, and ED21, n = 22), were used as non-operated controls and compared to the upper and lower segments of rat fetus small intestines with an induced atresia (n = 14; ligature at ED18). The pattern of gene expression was then confirmed by histochemistry, electron microscopy, and RT-qPCR.

### Results

From ED15 to ED21, the expression of several genes exhibited a physiological increase of ES markers, with a significant increase at the end of gestation. The operated embryos exhibited significantly higher variations of gene expression in the proximal segment than in the distal segment in terms of absorption and the epithelial barrier. An increase in goblet cells and markers was observed in the proximal segment compared to the controls.

### Conclusion

Fetal intestinal obstruction accelerates maturation in the proximal segment and disrupts the intestinal wall in the distal segment, with a decrease in the number of mucosal cells.

**Data Availability Statement:** All relevant data are within the manuscript and Supporting Information files. Microarray analysis of this study was performed at the genomic platform (GENOMIC'S) of the Institut Cochin in Paris (22 Rue Mechain,

75014 Paris) and it has been assigned accession number EMTAB-5981 in ArrayExpress. The files of the microarray analysis are publicly accessible in ArrayExpress. The accession number is: E-MTAB-5981 DOI:https://www.ebi.ac.uk/arrayexpress/experiments/E-MTAB-5981/.

**Funding:** This work was supported by a grant from CHU de Limoges, APREL (Appel à projet Recherche Equipes Emergentes et Labellisées, 2013), funded by ARS Limousin. The funders had no role in study design, data collection and analysis, decision to publish, or preparation of the manuscript.

**Competing interests:** The authors have declared that no competing interests exist.

Moreover, the epithelial cells underwent significant changes, supporting the notion that intestinal disorders involve more than the ENS.

## Introduction

Congenital intestinal atresia is a well-known cause of neonatal obstruction, with an estimated overall incidence of 1:1,500–1:2,000 live births [1, 2]. In jejunoileal atresia, the obstruction is thought to be related to a fetal vascular event secondary to mesenteric ischemia in the second or third trimester of pregnancy [3]. Treatment consists of surgical repair shortly after birth. The main postoperative complications affecting one-third of all cases include intestinal dysmotility and bacterial translocation, which may be life-threatening in severe cases.

The underlying pathophysiology of intestinal motility disorders is considered to be mainly related to enteric nervous system impairment, and most of the recent studies with animal models have focused on enteric nervous system changes [4–6]. Histological assessment of specimens from patients with intestinal atresia has confirmed the presence of an impaired ENS structure in the proximal segment and delayed maturation in the distal segment [7].

The underlying intestinal impairment in the postoperative period is, however, not fully understood. Enteroendocrine cells have recently been implicated in this pathological condition [8]. Altered enteroendocrine development can lead to impaired glucose homeostasis [9]. These results also suggest that other mucosal components such as glandular secretion or absorption may be disrupted in this intestinal obstruction [10]. Epithelial functions include the digestion and absorption of nutrients, the formation of an intestinal barrier against mechanical or biological stresses, and immune protection.

Early intestinal impairments such as childhood enteropathies with severe diarrhea can be associated with epithelial abnormalities [11]. In inflammatory bowel diseases, defects in intestinal barrier function such as a defective mucus layer [12] allow bacteria to come in direct contact with the epithelium, thereby enhancing bacterial invasion. Perturbation of epithelial permeability occurs in the same conditions, although the underlying molecular mechanisms are not fully understood [12]. Such considerations are not available for congenital conditions. Pathological epithelial changes have previously been highlighted in gastroschisis, which is another congenital condition involving intestinal obstruction. Perturbation of epithelial permeability with impairment of nutrient uptake has been found to be involved [13, 14]. Morphological assessment of the gut wall in atresia animal models has revealed a modified histological pattern in the proximal and distal segments close to the obstruction [15]. Other studies, of human samples, have reported the presence of differentiation of the epithelial wall at both sides of the obstruction [16, 17].

The aim of our study was to determine whether the proximal or the distal segment was the most impaired and to analyze the epithelial changes, in light of their suspected involvement in digestion impairment and bacterial translocation. To do so, a transcriptomic analysis was performed to identify changes in gene expression during late normal intestinal development (between ED15 and ED21) and in a rat model of surgically-induced intestinal obstruction (ligature of the small intestine in rat embryos at ED18) [18]. The gene expression pattern was assessed by RT-qPCR, histochemistry, and electron microscopy.

## Materials and methods

### 1. Ethics statement

The experimental protocol was approved by the French animal care and use committee (Comité National de Réflexion Ethique sur l'Expérimentation Animale; reference number 05180.04) and

all animal procedures were performed according to the French guidelines for animal protection and welfare.

## 2. Fetal bowels and intestinal obstruction

Pregnant Wistar rats were purchased from Janvier Labs (Le Genest-Saint-Isle, France). Pregnant rats were deeply anesthetized by inhalation of 3% isoflurane in combination with a mixture of nitrous oxide and oxygen (1:2, V/V). Kinetic control bowels at different developmental stages (ED15, ED17, ED19, and ED21) were obtained after cesarean section and fetus laparotomy.

Intestinal obstruction was surgically induced at day ED18 in fetuses of pregnant Wistar rats [9]. We selected well-exposed fetuses. The fetal abdominal wall was opened, and an intestinal loop was extracted without causing liver or umbilical cord injury. A 10/0 Prolene (Ethicon, France) suture was made around the exposed intestine. The abdominal loop was reintegrated into the abdominal cavity and then closed. During the same procedure, simple laparotomy was performed in other fetuses that were used as sham controls. At day ED21, and to prevent cannibalism, all of the animals were delivered by cesarean section.

The studied fetuses were weighed and measured. Their bowels were collected and measured. The criteria for successful surgery were: (i) a live fetus with a precise gestational age and (ii) macroscopic evidence of intestinal obstruction, i.e., a proximal intestine filled with biliary fluid and dilation of at least 2 cm distinct from a consistently uncolored distal segment. The obstructed bowels were analyzed on both sides of the ligature.

## 3. RNA isolation

Total RNA was extracted from bowels using QIAzol® Reagent and then treated with DNase I prior to purification using a Qiagen RNeasy Mini Kit according to the manufacturer's instructions (Qiagen, Hilden, Germany). The concentration and the purity of the eluted RNA were analyzed by spectrophotometry using a NanoDrop® ND-2000/2000c (Thermo Fisher Scientific, Waltham, Massachusetts, USA). The RNA quality and integrity were assessed with an Agilent 2100 bioanalyzer using an RNA 6000 Nano Kit (Agilent Technologies, Santa Clara, CA, USA). Only RNA samples with an RNA integrity number (RIN) greater than 7 were used. The samples were stored at -80 ˚C until use.

## 4. Microarray screening and data analysis

An Affymetrix GeneChip® Rat Gene 2.0 Array was used to analyze differential gene expression profiles of the control intestines at various stages of maturation (between ED15 and ED21, n = 16) and the segments proximal and distal to intestinal obstruction (ED21, n = 8). The samples were prepared according to the manufacturer's instructions and recommendations (Affymetrix, Santa Clara, CA, USA). Briefly, total RNA from each sample was reverse-transcribed, converted into complementary RNA using the standard Affymetrix protocol and hybridized to the Affymetrix GeneChip Rat Gene 2.0 Array at the genomic platform at the Cochin Institute.

**Data processing and analysis.** Raw Affymetrix data (.cel files) from the arrays were transformed by the Robust Multiarray Analysis (RMA) method using Bioconductor in R software to obtain genome-level expression values. The probe-level raw intensities were background-corrected using RMA, and then quantile normalized and summarized as log-expression values.

**Statistical analysis.** Normalized scanned probe array data were compared between groups to generate p-values and signal log-ratios (fold changes). The changes in gene expression were analyzed using unsupervised hierarchical clustering (with log-transformed and

normalized signal intensity values) and principal component analysis (PCA) to assess the data in terms of technical bias and outlier samples, and to visualize intergroup differences. To identify differentially expressed genes, an analysis of variance (ANOVA) was used for each gene and pairwise Tukey's post-hoc tests were used for comparisons between groups. Differences were considered significant when $p < 0.05$.

## 5. Quantitative real-time PCR

Complementary DNA was obtained by reverse transcription (RT) of the extracted total RNA (0.5 µg) with a QuantiTec Reverse Transcription Kit following the manufacturer's instructions (Qiagen, Hilden, Germany). Two RTs were performed for each RNA sample. All of the cDNA samples were quantified using a Nanodrop ND-2000/2000c spectrophotometer (Thermo, Waltham, Massachusetts, USA) to determine the concentration and purity of each cDNA sample. Primers for the genes of interest and the reference genes were designed using PrimerBlast (NCBI) and Primer3 software, with melting temperatures ranging from 62 ˚C to 63 ˚C. The sequences of the forward and the reverse primers used for the real-time PCR amplifications are shown in Table 1. The cDNA was amplified by real-time PCR using a RotorGene SYBR Green PCR Kit (Qiagen, Hilden, Germany) in a total volume of 25 µL consisting of 1x RotorGene SYBR Green MasterMix, 600 nM of each primer, and 80 ng of cDNA. The amplification was carried out using a Rotor-Gene 6000 multiplex system (Corbett Research) with the following qPCR conditions: 95 ˚C for 10 min, followed by 40 cycles of 95 ˚C for 30 s, and then 62 ˚C for 45 s. Each cDNA sample analysis was performed in duplicate for the genes of interest and in triplicate for the reference genes. Negative (water) controls were included in all of the qPCR runs.

The data were analyzed using the corresponding software version 1.7 (Corbett Research). Gene expression was quantified using the relative delta Ct method by comparing the number of cycles of target genes with the internal reference genes (HPRT and TBP). Target gene/reference gene ratios were calculated and expressed as $2^{-\Delta Ct}$. This ratio was then used to evaluate the expression level of the target genes in each sample.

The data were analyzed in duplicates of three independent samples (mean ± SD) of three groups: proximal, distal, and control intestines at ED21. Differences in gene expression were analyzed by quantitative RT-PCR using the Student's *t*-test. Differences were considered to be significant at $p < 0.05$.

**Table 1. Primer sequences for the real-time PCR.**

| Genes | Primer sequences 5'-3' | | |
|---|---|---|---|
| | Forward (F)/Reverse (R) primers | | |
| TMPRSS15 | F: | TGACTGGCTGGTGTCTGCTG |
| | R: | CGATCGACCACCCGTCTTACT |
| LCT | F: | CCCAAGGGCTTCATCTGGAGT |
| | R: | CATGCCACGTCTCCGTTGTC |
| TFF3 | F: | CTGGATAACCCTGCTGCTGGT |
| | R: | GCCGGGACCATACATTGGCT |
| HPRT | F: | CTCATGGACTGATTATGGACAGGAC |
| | R: | GCAGGTCAGCAAAGAACTTATAGCC |
| TBP | F: | TGGGATTGTACCACAGCTCCA |
| | R: | CTCATGATGACTGCAGCAAACC |

Abbreviations: TMPRSS15: Transmembrane Serine Protease Serine 15, LCT: Lactase, TFF3: Trefoil Factor 3, TBP: TATA box binding protein, HPRT: Hypoxanthine-guanine phosphoribosyltransferase.

## 6. Optical microscopy

Bowels were collected, oriented, and fixed immediately on ice in 4% paraformaldehyde (PFA) in PBS (phosphate-buffered saline solution, pH 7.4). After 3 hours, the PFA was removed, and the bowels were rinsed with PBS and incubated in 30% (w/v) sucrose in PBS overnight at 4 ˚C. The samples were frozen by immersion in liquid nitrogen-cooled N-methyl butane. The proximal and distal segments on both sides of the atresia were embedded together from the ligature. Sections with a thickness of 8 µm were cut using a freezing microtome and processed for immunohistochemistry or histology. For histology, the sections were stained with hematoxylin and eosin to visualize the overall morphology.

Sections of intestines were stained with periodic acid-Schiff reaction (PAS) to detect intestinal goblet cells. The number of PAS+ goblet cells was expressed per section.

For morphometry, quantification of the absorption area was calculated using a surface modeling pattern based on measurement of the number of villi, as well as the length and the intestinal diameter. The total absorption area was calculated using the total area of villi. Differences in the area between the proximal and the distal segment were analyzed quantitatively using the Student's *t*-test. Differences were considered to be significant at $p < 0.05$.

## 7. Transmission electron microscopy (TEM)

The samples were fixed in 2.5% glutaraldehyde in 0.05 mol/L sodium cacodylate buffer, post-fixed in 1% osmium tetroxide, and embedded in epoxy resin. Semi-thin sections (1 µm thickness) were stained with toluidine blue and examined under a light microscope. Areas appropriate for further ultrathin sectioning were then selected and examined using a transmission electron microscope. Ultrathin sections were stained with uranyl acetate and lead citrate, and then examined with an electron microscope (a JEM-1011, JEOL, Tokyo, Japan) at 80 KeV. Each obstructed bowel was compared to a control bowel in order to analyze the epithelium ultrastructures.

## Results

### 1. Analysis of global gene expression

Microarray gene expression was studied on proximal and distal segments of bowel ligatures (n = 4) and on control bowels at different stages of differentiation ED15 (n = 4), ED17 (n = 4), ED19 (n = 4), and E21 (n = 4). Compared to ED21 controls, 3,625 genes were significantly up- or down-regulated in the segment proximal to the ligature (18% of the 20,267 genes) whereas only half of the genes (1,894 genes, 9%) exhibited significant changes in the distal segment (Table 3). The results show that global gene expression was more disrupted in the proximal segment than in the distal segment compared with E21 segment controls (Table 2). Thus, the proximal segments appeared to be more disrupted by the ligature.

**Table 2. Significantly up- or down-regulated gene numbers at various stages of development for kinetic controls and in the obstructed bowel.**

|  | Controls E17 vs. E15 | Controls E19 vs. 15 | Controls E21 vs. E15 | Proximal vs. E21 controls | Distal vs. E21 controls |
|---|---|---|---|---|---|
| Genes up- or down regulated, p<0.05 | 3,853 | 8,666 | 12,315 | 3,625 | 1,894 |
| Genes up-regulated, p<0.05 | 2,046 | 3,554 | 4,338 | 1,452 | 743 |
| Genes down-regulated, p<0.05 | 1,807 | 5,112 | 7,977 | 2,173 | 1,151 |

**Table 3. Morphometric results for the intestinal absorption area.**

| Absorption area data | E21 proximal (n = 3) | | E21 distal (n = 3) | | E21 Control (n = 3) |
|---|---|---|---|---|---|
| | Mean ± SD | p-value vs. controls | Mean ± SD | p-value vs. controls | Mean ± SD |
| **Measured data** | | | | | |
| Intestinal diameter (μm) | 1,245.5 ± 113.03 | 0.049 | 629.13 ± 88.38 | 0.012 | 902.97 ± 46.88 |
| Villi length (μm) | 299.73 ± 83.51 | 0.512 | 214.47 ± 35.62 | 0.062 | 267.28 ± 31.98 |
| Villi numbers | 37.27 ± 3.91 | 0.045 | 18.13 ± 0.81 | 0.025 | 22.53 ± 0.50 |
| **Calculated data** | | | | | |
| Villi diameter (μm) | 98.15 ± 6.55 | 0.080 | 102.40 ± 12.25 | 0.088 | 120.19 ± 4.07 |
| Villi area (μm²) | 93,400 ± 31,000 | 0.091 | 70,700 ± 17,900 | 0.047 | 101,000 ± 12,500 |
| Intestinal area (mm²) | 38.63 ± 9.74 | 0.048 | 14.31 ± 2.79 | 0.027 | 21.66 ± 2.7017 |

## 2. Surface area of the small intestine

**Morphometry and optical microscopy.** Calculation of the bowel absorption area revealed a significant statistical increase in the proximal segment ($38.6 \text{ mm}^2 \pm 9.7$ vs. $21.63 \text{ mm}^2 \pm 2.7$, p = 0.04) and a decrease in the distal segment ($14.3 \text{ mm}^2 \pm 2.8$ vs. $21.6 \text{ mm}^2 \pm 2.7$, p = 0.02) (Table 3).

## 3. Enterocyte cell brush-border membrane transporters and enzymes

**Gene expression.** A strong increase in gene expression for brush-border membrane transporters and enzymes, in particular for lactase, was observed in the last days of normal fetal development between ED15 and ED21 (S1 Table). No significant differences in gene expression were observed in atretic fetuses in the proximal and the distal segments compared to the ED21 controls.

Lactase gene expression increased significantly in the proximal and the distal segments compared to the ED21 controls based on quantitative real-time PCR (Table 4). Based on quantitative real-time PCR, enterokinase expression (TMPRSS15) increased in the proximal segment (Table 4 and S1 Table).

**Electron microscopy.** The ED18 enterocytes exhibited both a reduced number and shorter microvilli than the ED21 controls (Fig 1). In the ED21 proximal and distal segments, the length of the microvilli was shorter (Fig 1). The epithelium ultrastructure of the distal segments revealed two types of enterocytes: (1) enterocytes that had a dark cytoplasm and an ovoid nucleus at the basal third and (2) enterocytes that had a clear cytoplasm and a large basal nucleus (Fig 1). These two populations of enterocytes may correspond to two different stages of maturation. Tight junctions between enterocytes were present at ED18 and ED21. They were not modified in the proximal or the distal segment.

**Table 4. Quantitative real-time PCR results: Gene expression and p-values of selected brush-border membrane enzymes and transporters.**

| Genes | E21 proximal (n = 3) | | | | E21 distal (n = 3) | | | | E21 Control (n = 3) | | | p-value |
|---|---|---|---|---|---|---|---|---|---|---|---|---|
| | Mean | ± | SD | p-value vs. controls | Mean | ± | SD | p-value vs. controls | Mean | ± | SD | Proximal vs. Distal |
| **BRUSH BORDER MEMBRANE ENZYMES AND RANSPORTERS** | | | | | | | | | | | | |
| TMPRSS15 | 8.97 | ± | 11.66 | 0.254 | 0.02 | ± | 0.01 | 0.175 | 0.01 | ± | 0.01 | 0.254 |
| LCT | 30.56 | ± | 4.48 | 0.006 | 37.59 | ± | 11.39 | 0.018 | 9.36 | ± | 5.26 | 0.376 |

Abbreviations: TMPRSS15: Transmembrane Serine Protease Serine 15, LCT: Lactase,

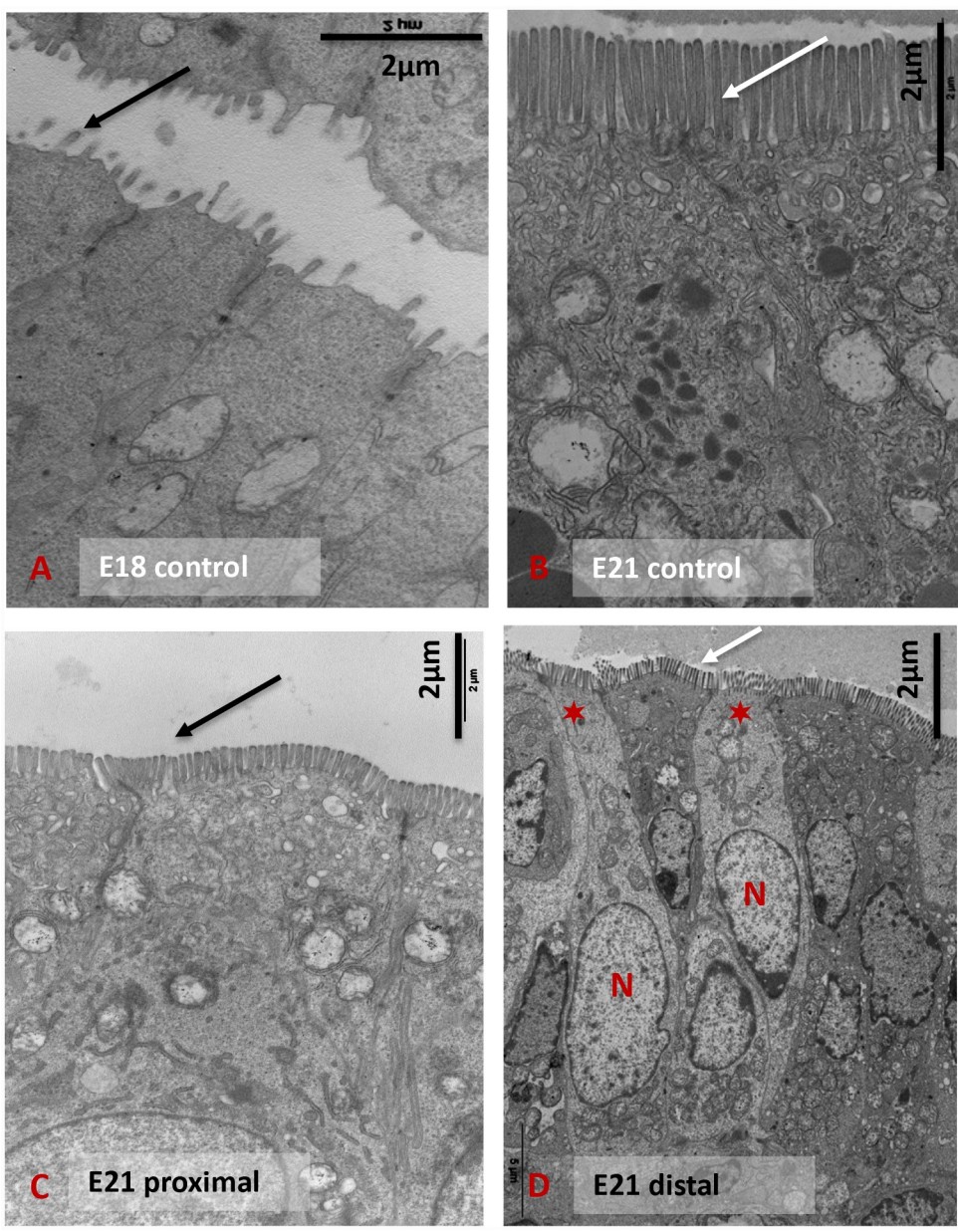

**Fig 1. The ultrastructure of enterocytes.** Electron microscopy focusing on enterocytes with microvilli on the apical surface of E18 controls (A), E21 controls (B), E21 proximal segment (C), and E21 distal segment (D). N indicates cell nuclei; ✳ indicate immature observed enterocytes and arrows indicate microvilli.

## 4. Goblet cells and mucins

**Histochemistry (PAS).** Goblet cells were present in the epithelium at the ED21 developmental stage but absent at ED18. The number of goblet cells was significantly increased in the proximal segment compared with the ED21 controls ($121.3 \pm 16.3$ vs. $69.7$ mm$^2$ $\pm 13.6$; $p = 0.01$) (Table 5).

**Gene expression.** As for enterocytes, an increase in gene expression for a number of goblet cell markers was observed in the last days of fetal development between ED15 and ED21

**Table 5. Goblet cells in the epithelium at ED21.** The number of goblet cells was significantly increased in the proximal segment compared with the ED21 controls.

| | E21 proximal (n = 3) | | E21 distal (n = 3) | | E21 Control (n = 3) | p-value |
|---|---|---|---|---|---|---|
| **PAS+ cells per section** | **Mean ± SD** | **p-value vs. controls** | **Mean ± SD** | **p-value vs. controls** | **Mean ± SD** | **Proximal vs. Distal** |
| | 121.33 ± 16.29 | 0.012 | 69.67 ± 13.65 | 0.340 | 78.67 ± 4.62 | 0.014 |

(S1 Table). This increase occurred both for membrane-bound and for secreted mucins. Significant increases in mucins were observed in the proximal segment of atretic fetuses, particularly for the secreted mucin 5b (S1 Table).

**Electron microscopy.** No goblet cells were observed at the ED18 developmental stage. The goblet cells of the proximal and the distal segments exhibited similar structures compared to ED21 (Fig 2).

## Discussion

This article documents a new approach to study prenatal epithelial behavior after intestinal obstruction. It confirms the presence of all epithelial intestinal markers at delivery in this rat model. The results show that obstruction affects the proximal and the distal segment, with an overall increase in epithelial maturation and an increase in the number of goblet cells in the proximal segment associated with an increase in mucins secretion and the intestinal area compared to controls. In terms of global gene expression, the effect of the ligature was more pronounced in the proximal segment. Overall accelerated maturation has previously been reported in the same model [8], particularly for endocrine cells. The same observation for epithelial cells appears to be relevant as both share the same bowel location and endodermal embryological cell precursors. In the distal segments, the gene expression pattern was less altered compared to the controls. According to our findings, the proximal segment appeared to be more affected by the ligature, given the significantly higher number of genes that were differentially expressed based on analysis of the global transcriptome. The gene expression profile confirmed the accelerated maturation of this segment, as previously described [7]. This assumption is supported by the mechanical theory in the developmental process [19]. This theory is based on evidence that mechanical stresses induce mechanotransduction pathways that

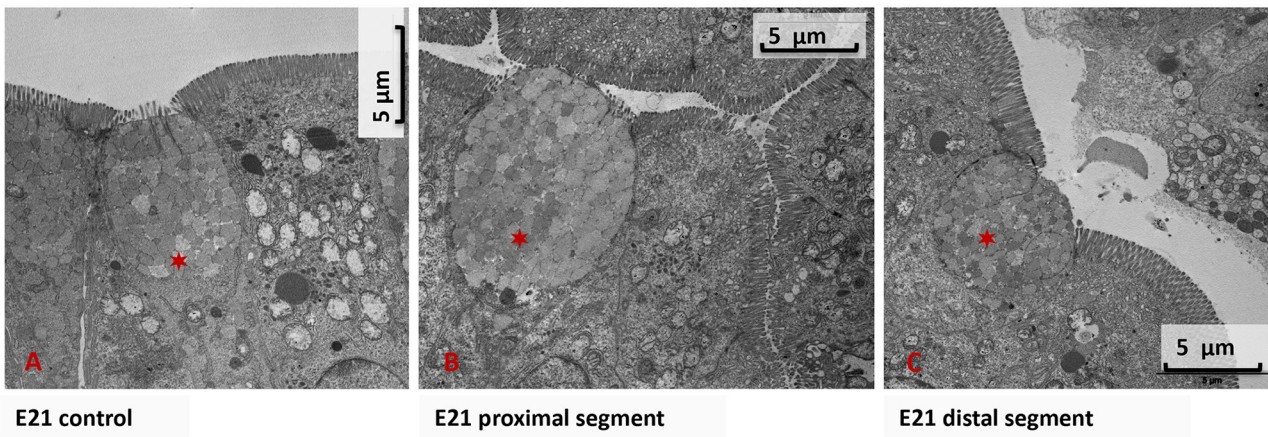

|  E21 control  |  E21 proximal segment  |  E21 distal segment  |

**Fig 2. The ultrastructure of goblet cells: A, B, and C represent epithelium with goblet cells indicated by ∗.** A corresponds to E21 controls, B to E21 proximal segment, and C to E21 distal segment.

then lead to epithelial cell proliferation and tissue differentiation. The changes in luminal fluid pressure and the interruption of the flow of amniotic fluid induced by the ligature could impact the development of the proximal segment but not that of the distal segment.

Intestinal absorption appears to be enhanced in the proximal segment due to an increase in the secretion of enzymes and an increase in the area. The bowel area has been estimated in the literature based on the external diameter of the bowel [6, 18]. In the present study, a dedicated model was implemented that included the length of the villi in the calculation of this parameter. In terms of the secretion of enzymes, there has been little information to date in the literature. To the best of our knowledge, only one article has examined the epithelium and intestinal atresia [16]. Various markers (enterocytes, endocrine, goblet, and Paneth cells) were described in the proximal and the distal segment of the atresia of human newborns using immunohistochemical techniques. No changes in organization such as perturbation of the balance between proliferation and differentiation, comprising hypertrophy or atrophy, were reported at either side of the atresia.

Gastrointestinal mucins are glycoproteins produced by goblet cells. They comprise the main structural components of the mucus layer. Membrane-bound and secreted mucins play a critical role in regulatory responses against microorganisms [20]. They have been studied extensively in the context of inflammatory bowel disease [21], but there have been very few reports of their involvement in intestinal obstruction (S2 Table). Abnormal mucin secretion can lead to enterocolitis in neonates with intestinal congenital obstruction [22]. Here, a significant increase in goblet cells and associated specialized mucins in the proximal segment may involve a thicker mucus layer as a first line of defense against pathogenic microorganisms [23]. Goblet cells appeared between E19 and E21, after the surgical ligature in our model. In our opinion, the absence of an increase in mucus in the distal segment may be explained by the lack of mechanical or nutritional stimulation. Our results are not in accordance with those of Schaart et al. [16], who found no histological differences between the proximal and distal segments. However, their results need to be interpreted with a degree of caution as they only assessed mucin 2 and TTF3, without providing any details regarding the distance between the sample and the atresia site, and they did not use a quantification method.

The present results support the notion of increased wall thickness and a barrier effect in the proximal segment and less of a barrier in the distal wall. Intestinal defense and permeability may be also influenced by other parameters such as the immune system and inflammation, which were not studied here. No ultrastructural modifications of the tight junctions could be observed, but functional considerations are lacking. Khen et al. [6] reported enhanced permeability at both sides of the atresia using functional studies based on the same rat model. These authors did not find any difference between the two segments in terms of inflammation. The limitation of our study is the lack of functional experiments, particularly for enzyme digestion. It would be interesting to explore the changes in epithelial parameters after surgical treatment of the intestinal obstruction. A study using a rat model has concluded that the enteric nervous system has a high regenerative capacity after anastomoses [24]. Confirmation of this experimental study with human newborns would be complicated due to the very limited availability of postoperative bowel samples, particularly of the distal segment, with potential ethical concerns.

These results support the description of a proximal enhanced maturation segment with an increased absorption area and mucus layer, while the distal segment is not stimulated. Reduction in the number of goblet and Paneth markers and cells has been reported to be significantly associated with bacterial infection in the case of necrotizing enterocolitis in human newborns [25]. In the case of intestinal atresia, it is thought that changes in transcellular transport provide pathophysiological support to gut-derived sepsis in the postoperative period. Hence, our findings are in agreement with the mechanisms of the potential occurrence of bacterial

translocation and they suggest involvement of the distal segment after surgical treatment of intestinal atresia in humans.

## Conclusion

Intestinal obstruction impairs epithelial gut development, mainly in the proximal segment, and it accelerates maturation. This segment probably undergoes a pathological increase in the intraluminal pressure due to the interrupted flow of amniotic fluid. In contrast, the distal segment of the atretic intestine is not significantly affected, with a decrease in the absorption area and mucus components. Modification of the intestinal barrier may explain bacterial translocations, with consequent severe gut-related sepsis after surgical treatment of intestinal atresia. Further research is needed to confirm these findings and to determine the influence of inflammation and the immune system on epithelial permeability.

## Supporting information

**S1 Table. Transcriptomic results of epithelial components.** Transcriptome results: fold-changes and p-values for the gene expression of brush-border membrane enzymes, transporters, and mucins expressed in the duodenum and the small intestine of the control fetuses at ED17 vs. ED15, ED19 vs. ED15, ED21 vs. ED15 (n = 4) and ED21 (n = 4) proximal and distal segments compared to the ED21 controls (n = 4).
(XLS)

**S2 Table. Review of the epithelial changes in intestinal atresia.** Abbreviations: TEM: transmission electron microscopy; IHC: immunohistochemistry; HE: hematoxylin and eosin staining; HES: hematoxylin, eosin, Saffron staining; ED: embryonic development.
(XLS)

## Acknowledgments

We wish to thank Angelique Nizou for the logistic support and the Cochin Institute for technical support with the transcriptomic analysis.

## Author Contributions

**Conceptualization:** Quentin Ballouhey, Laurent Fourcade, Franck Sturtz, Sylvie Bourthoumieu.

**Data curation:** Quentin Ballouhey, Camille Bellet, Chaharazed El Hamel, Jean Michel Vallat, Franck Sturtz.

**Formal analysis:** Chaharazed El Hamel, Sylvie Bourthoumieu.

**Funding acquisition:** Laurence Richard, Franck Sturtz, Sylvie Bourthoumieu.

**Investigation:** Quentin Ballouhey, Laurent Fourcade, Laurence Richard, Camille Bellet, Franck Sturtz, Sylvie Bourthoumieu.

**Methodology:** Quentin Ballouhey, Laurent Fourcade, Jean Michel Vallat, Franck Sturtz, Sylvie Bourthoumieu.

**Project administration:** Sylvie Bourthoumieu.

**Resources:** Quentin Ballouhey, Laurent Fourcade, Laurence Richard, Camille Bellet, Jean Michel Vallat, Sylvie Bourthoumieu.

**Software:** Chaharazed El Hamel, Jean Michel Vallat.

**Supervision:** Franck Sturtz, Sylvie Bourthoumieu.

**Validation:** Quentin Ballouhey, Jean Michel Vallat, Franck Sturtz, Sylvie Bourthoumieu.

**Visualization:** Quentin Ballouhey, Franck Sturtz, Sylvie Bourthoumieu.

**Writing – original draft:** Quentin Ballouhey, Franck Sturtz, Sylvie Bourthoumieu.

**Writing – review & editing:** Franck Sturtz, Sylvie Bourthoumieu.

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
