## [Decision Letter · Decision Letter 0]

2 Jan 2020

PONE-D-19-26433

Epithelial changes of congenital intestinal obstruction in a rat model

PLOS ONE

Dear Dr Ballouhey,

Thank you for submitting your manuscript to PLOS ONE. After careful consideration, we feel that it has merit but does not fully meet PLOS ONE’s publication criteria as it currently stands. Therefore, we invite you to submit a revised version of the manuscript that addresses the points raised during the review process.

We would appreciate receiving your revised manuscript by Feb 16 2020 11:59PM. To enhance the reproducibility of your results, we recommend that if applicable you deposit your laboratory protocols in protocols.io, where a protocol can be assigned its own identifier (DOI) such that it can be cited independently in the future. For instructions see: http://journals.plos.org/plosone/s/submission-guidelines#loc-laboratory-protocols

We look forward to receiving your revised manuscript.

Kind regards,

Shree Ram Singh, Ph. D.

Academic Editor

PLOS ONE

Journal Requirements:

3.  Thank you for including the following funding information within the acknowledgements section of your manuscript; "This work was supported by a grant from the UHC of Limoges, APREL (Appel à projet Recherche Equipes Emergentes et Labellisées, 2013), funded by the ARS Limousin."

Reviewers' comments:

Reviewer's Responses to Questions

**Comments to the Author**

1. Is the manuscript technically sound, and do the data support the conclusions?

Reviewer #1: Yes

Reviewer #2: No

2. Has the statistical analysis been performed appropriately and rigorously? 

Reviewer #1: Yes

Reviewer #2: No

3. Have the authors made all data underlying the findings in their manuscript fully available?

Reviewer #1: Yes

Reviewer #2: No

4. Is the manuscript presented in an intelligible fashion and written in standard English?

Reviewer #1: Yes

Reviewer #2: Yes

5. Review Comments to the Author

Reviewer #1: Manuscript appears to be well explained and contains all the statistically significant supporting data. It would be good to mention the number of animals used for each group, while explaining materials method (though it has been mentioned later in results). Please, carefully read the manuscript and prefer to avoid using repetitive words (like "impaired" used twice in a single sentence). Also avoid repetitive use of "the" at many places(like "The ratio of the target genes to the reference genes"). Rest seems to be fine.

Reviewer #2: The manuscript entitled “Epithelial changes of congenital intestinal obstruction in a rat model” by Ballouhey et.al has described the effect of surgically induced atresia in rat embryos. Different models of atresia have been reported, however, most of them focused on morphological changes. The authors tried to advance previous reports on morphological changes by taking a gene expression profile-based approach together with morphological examinations. However, in my opinion, the manuscript fails to address the specific scientific hypothesis. The conclusions, and the title of the manuscript are not supported by data.

Major concerns:

1. Overall representation of the manuscripts looks more incoherent. The result sections are very confusing and redundant.

2. Sample size for most of the data are very small (n=3). Previous studies by Fourcade et. al. had used n>10 in the rat model.

3. Table 2; The statistical power of this data is not so convincing considering the small sample size (n=3). It is difficult to assume the gaussian distribution of the data with the small sample size and with high variation. Therefore, I suggest non parametric test for this set of data.

4. Table 3: number of deferentially expressed gene does not make any scene unless the they suggest any biological function or pathway. I suggest a gene ontology analysis of the deferentially expressed gene to better represents the biological changes.

5. Is the gene expression data deposited in any data bank?

6. Figures should be more organized and properly labelled for the clarity of the readers.

Minor concerns

Please provide HPRT and TBP primer sequence in table 1.

6. PLOS authors have the option to publish the peer review history of their article (what does this mean?). If published, this will include your full peer review and any attached files.

Reviewer #1: No

Reviewer #2: No

---

## [Author Response · Author response to Decision Letter 0]

4 Mar 2020

Response to Reviewer 1

Reviewer 1: Manuscript appears to be well explained and contains all the statistically significant supporting data. It would be good to mention the number of animals used for each group, while explaining materials method (though it has been mentioned later in results). Please, carefully read the manuscript and prefer to avoid using repetitive words (like "impaired" used twice in a single sentence). Also avoid repetitive use of "the" at many places (like "The ratio of the target genes to the reference genes"). Rest seems to be fine.

We thank Reviewer 1 for their very useful comments. As suggested in the comments, we have changed the following:

-“Impaired” has been replaced with “Altered” in the introduction section

-The sentence “The ratio of the target genes to the reference genes was calculated and expressed as 2-ΔCt.” Was modified to “Target gene/reference gene ratios were calculated and expressed as 2-ΔCt.”

-The style of the entire manuscript style has also been checked.

Response to Reviewer 2

Reviewer 2: The manuscript entitled “Epithelial changes of congenital intestinal obstruction in a rat model” by Ballouhey et al. has described the effect of surgically induced atresia in rat embryos. Different models of atresia have been reported, however, most of them focused on morphological changes. The authors tried to advance previous reports on morphological changes by taking a gene expression profile-based approach together with morphological examinations. However, in my opinion, the manuscript fails to address the specific scientific hypothesis. The conclusions, and the title of the manuscript are not supported by data.

Response to Reviewer 2

1. Overall representation of the manuscripts looks more incoherent. The result sections are very confusing and redundant.

Author’s response: We apologize for the fact that Reviewer 2 found the Results section to be confusing. In light of this remark, we have changed the followings points in the Results paragraph:

-“Analysis of global gene expression” section has been moved in the first part of the paragraph.

-The presentation of the table illustrating the morphometric results has been modified. 

-Throughout the entire paragraph, “Microvilli” has been substituted with “Villi” as Microvilli can only been detected by electron microscopy.

-The “Enterocyte cell brush border” section has been modified and the last sentence has been deleted “Sucrase isomaltase…(S1 Table)”.

2. Sample size for most of the data are very small (n=3). Previous studies by Fourcade et al. had used n > 10 in the rat model.

Author’s response: Indeed, the sample size was n = 3 or = 4, as the guidelines for animal experimentation stipulate that as few animals as possible are used. The aim of the experiment was to use a variety of techniques rather than a large number of animals. The transcriptomic experiment required n = 3 for each group, which meant using more than 18 operated animals, not including the high mortality rate during the surgical procedures. In previous published studies, the number of animals was higher but only a single technique was used, for example Fourcade et al. only performed an optical microscopy analysis. 

3. Table 2: The statistical power of this data is not so convincing considering the small sample size (n=3). It is difficult to assume the gaussian distribution of the data with the small sample size and with high variation. Therefore, I suggest non parametric test for this set of data.

Author’s response: We fully agree with this remark. Consequently, we have opted to use a non-parametric test and we have made the following changes:

-The sentence corresponding to the statistical analysis has been changed in the Methods section: Differences in the area between the proximal and the distal segment have been analyzed quantitatively using the Wilcoxon test.

-The data in the Results section corresponding to Table 3 have been changed.

A qualified statistician has now reviewed the statistics of the manuscript and this individual has hence been added to the list of authors.

4. Table 3: number of deferentially expressed gene does not make any scene unless they suggest any biological function or pathway. I suggest a gene ontology analysis of the deferentially expressed gene to better represents the biological changes.

Author’s response: the objective was not to produce biologically significant results but to determine which segment was the most impaired. In accordance with the reviewer’s comment, we have made the following modification:

-The beginning of the Introduction section has changed to “The aim of our study was to determine whether the proximal or the distal segment was the most impaired and …to analyze…”

-The Results section has also been modified and the “Analysis of global gene expression” section along with the related Table (Table 3 becoming Table 2) has been moved in the first part of this paragraph.

5. Is the gene expression data deposited in any data bank?

Author’s response: Yes, the data are available online. The microarray analysis of this study was performed at the genomic platform (GENOMIC’S) of the Institut Cochin in Paris (22 Rue Mechain, 75014 Paris) and it has been assigned accession number E-MTAB-5981 in ArrayExpress.

6. Figures should be more organized and properly labelled for the clarity of the readers.

Author’s response: We apologize for the fact that Reviewer 2 found our pictures to be in want of improvement. In accordance to this, all of the patterning schematics of the figures have been changed to improve their appearance: the labelling of the Figures 1 and 2 has been modified. A legend has been added for each picture and the scale bars have been modified.

7. Minor concerns. Please provide HPRT and TBP primer sequence in table 1.

Author’s response: the HPRT and TBP primer sequences are now provided in Table 1.

---

## [Decision Letter · Decision Letter 1]

2 Apr 2020

PONE-D-19-26433R1

Epithelial changes of congenital intestinal obstruction in a rat model

PLOS ONE

Dear Dr Ballouhey,

Thank you for submitting your revised manuscript to PLOS ONE. After careful consideration, we feel that it has merit but does not fully meet PLOS ONE’s publication criteria as it currently stands. Therefore, we invite you to submit a revised version of the manuscript that addresses the minor points raised by a reviewer on your revised manuscript. 

We would appreciate receiving your revised manuscript by May 17 2020 11:59PM. To enhance the reproducibility of your results, we recommend that if applicable you deposit your laboratory protocols in protocols.io, where a protocol can be assigned its own identifier (DOI) such that it can be cited independently in the future. For instructions see: http://journals.plos.org/plosone/s/submission-guidelines#loc-laboratory-protocols

We look forward to receiving your revised manuscript.

Kind regards,

Shree Ram Singh, Ph. D.

Academic Editor

PLOS ONE

Reviewers' comments:

Reviewer's Responses to Questions

**Comments to the Author**

1. If the authors have adequately addressed your comments raised in a previous round of review and you feel that this manuscript is now acceptable for publication, you may indicate that here to bypass the “Comments to the Author” section, enter your conflict of interest statement in the “Confidential to Editor” section, and submit your "Accept" recommendation.

Reviewer #2: (No Response)

2. Is the manuscript technically sound, and do the data support the conclusions?

Reviewer #2: Yes

3. Has the statistical analysis been performed appropriately and rigorously? 

Reviewer #2: Yes

4. Have the authors made all data underlying the findings in their manuscript fully available?

Reviewer #2: Yes

5. Is the manuscript presented in an intelligible fashion and written in standard English?

Reviewer #2: Yes

6. Review Comments to the Author

Reviewer #2: The authors have significantly improved the revised manuscript. I would suggest that the following analysis to be included in the manuscript for better elucidation of the intestinal differentiation in control embryos. As the authors have performed microarray with different stages of control samples, namely ED15, ED17, ED19 and E21, it is important to show differential gene expression of all three stages (i.e. ED17,19, 21) compare to ED15 and associated pathways instead of just number of genes in ED21 vs ED15 (table 3).

7. PLOS authors have the option to publish the peer review history of their article (what does this mean?). If published, this will include your full peer review and any attached files.

Reviewer #2: No

---

## [Author Response · Author response to Decision Letter 1]

6 Apr 2020

Response to Reviewer 2

Reviewer 2: Manuscript appears to be well explained and contains all the The authors have significantly improved the revised manuscript. I would suggest that the following analysis to be included in the manuscript for better elucidation of the intestinal differentiation in control embryos. As the authors have performed microarray with different stages of control samples, namely ED15, ED17, ED19 and E21, it is important to show differential gene expression of all three stages (i.e. ED17,19, 21) compare to ED15 and associated pathways instead of just number of genes in ED21 vs ED15 (table 2).

Author’s response: 

We thank Reviewer 2 for their very useful comments. 

We fully agree with this remark. 

Consequently, we have implemented the intermediate results of microarray E15-17-E19-E21 for kinetic controls in Table 2. 

We have implemented the intermediate results of microarray E15-17-E19-E21 kinetic controls for specific pathways in Table S1.

---

## [Editor Report · Decision Letter 2]

7 Apr 2020

Epithelial changes of congenital intestinal obstruction in a rat model

PONE-D-19-26433R2

Dear Dr. Ballouhey,

We are pleased to inform you that your revised  manuscript has been judged scientifically suitable for publication and will be formally accepted for publication once it complies with all outstanding technical requirements.

With kind regards,

Shree Ram Singh, Ph. D.

Academic Editor

PLOS ONE
---

## [Editor Report · Acceptance letter]

8 Apr 2020

PONE-D-19-26433R2 

Epithelial changes of congenital intestinal obstruction in a rat model 

Dear Dr. Ballouhey:

I am pleased to inform you that your manuscript has been deemed suitable for publication in PLOS ONE. Congratulations! Your manuscript is now with our production department. 

With kind regards,

on behalf of

Dr. Shree Ram Singh 

Academic Editor

PLOS ONE